A reappraisal of the Middle Triassic chirotheriid Chirotherium ibericus Navás, 1906 (Iberian Range NE Spain), with comments on the Triassic tetrapod track biochronology of the Iberian Peninsula

Díaz-Martínez Ignacio 1 inaportu@hotmail.com
Castanera Diego 2
Gasca José Manuel 2
Canudo José Ignacio 2
1 CONICET—Instituto de Investigación en Paleobiología y Geología, Universidad Nacional de Río Negro , General Roca, Río Negro , Argentina
2 Grupo Aragosaurus-IUCA, Área de Paleontología, Facultad de Ciencias, Universidad de Zaragoza , Zaragoza , Spain
De Baets Kenneth
Electronic publication date: 2015 Jun 23
Publication date: 2015
Volume: 3
Electronic Location ID: e1044
Received 2015 Mar 25; Accepted 2015 Jun 1
Copyright: © 2015 Díaz-Martínez et al.
Copyright year: 2015
Copyright holder: Díaz-Martínez et al.
License: This is an open access article distributed under the terms of the Creative Commons Attribution License, which permits unrestricted use, distribution, reproduction and adaptation in any medium and for any purpose provided that it is properly attributed. For attribution, the original author(s), title, publication source (PeerJ) and either DOI or URL of the article must be cited.
License URL: https://creativecommons.org/licenses/by/4.0/

Keywords: Chirotherium ibericus, Chirotherium barthii, Middle Triassic, Iberian Peninsula, Triassic geochronology, Vertebrate tracks

Funding: Group IT834-13 of the Basque Government Spanish Ministerio de Economía y Competitividad (MINECO the Government of Aragón CGL2010–16447 CGL2010–18851/BTE CGL2013–47521-P University of Zaragoza 221-393 Ministerio de Ciencia, Tecnología e Innovación Productiva Consejo Nacional de Investigaciones Científicas y Técnicas (CONICET) This research was supported by the group IT834-13 of the Basque Government, the projects CGL2010–16447, CGL2010–18851/BTE and CGL2013–47521-P of the Spanish Ministerio de Economía y Competitividad (MINECO the Government of Aragón) (“Grupos Consolidados” H54, and “Dirección General de Patrimonio Cultural”) and the University of Zaragoza (221-393). Ignacio Díaz Martínez is supported by a postdoctoral grant from the Ministerio de Ciencia, Tecnología e Innovación Productiva Consejo Nacional de Investigaciones Científicas y Técnicas (CONICET) of the Government of Argentina. The funders had no role in study design, data collection and analysis, decision to publish, or preparation of the manuscript.

==============================
Triassic vertebrate tracks are known from the beginning of the 19th century and have a worldwide distribution. Several Triassic track ichnoassemblages and ichnotaxa have a restricted stratigraphic range and are useful in biochronology and biostratigraphy. The record of Triassic tracks in the Iberian Peninsula has gone almost unnoticed although more than 25 localities have been described since 1897. In one of these localities, the naturalist Longinos Navás described the ichnotaxon Chirotherium ibericus in 1906.The vertebrate tracks are in two sandy slabs from the Anisian (Middle Triassic) of the Moncayo massif (Zaragoza, Spain). In a recent revision, new, previously undescribed vertebrate tracks have been identified. The tracks considered to be C. ibericus as well as other tracks with the same morphology from both slabs have been classified as Chirotherium barthii. The rest of the tracks have been assigned to Chirotheriidae indet., Rhynchosauroides isp. and undetermined material. This new identification of C. barthii at the Navás site adds new data to the Iberian record of this ichnotaxon, which is characterized by the small size of the tracks when compared with the main occurrences of this ichnotaxon elsewhere. As at the Navás tracksite, the Anisian C. barthii-Rhynchosauroides ichnoassemblage has been found in other coeval localities in Iberia and worldwide. This ichnoassemblage belongs to the upper Olenekian-lower Anisian interval according to previous biochronological proposals. Analysis of the Triassic Iberian record of tetrapod tracks is uneven in terms of abundance over time. From the earliest Triassic to the latest Lower Triassic the record is very scarce, with Rhynchosauroides being the only known ichnotaxon. Rhynchosauroides covers a wide temporal range and gives poor information for biochronology. The record from the uppermost Lower Triassic to the Middle Triassic is abundant. The highest ichnodiversity has been reported for the Anisian with an assemblage composed of Dicynodontipus, Procolophonichnium, Rhynchosauroides, Rotodactylus, Chirotherium, Isochirotherium, Coelurosaurichnus and Paratrisauropus. The Iberian track record from the Anisian is coherent with the global biochronology proposed for Triassic tetrapod tracks. Nevertheless, the scarcity of track occurrences during the late Olenekian and Ladinian prevents analysis of the corresponding biochrons. Finally, although the Iberian record for the Upper Triassic is not abundant, the presence of Eubrontes, Anchisauripus and probably Brachychirotherium is coherent with the global track biochronology as well. Thus, the Triassic track record in the Iberian Peninsula matches the expected record for this age on the basis of a global biochronological approach, supporting the idea that vertebrate Triassic tracks are a useful tool in biochronology.

Introduction

Vertebrate track morphology is mainly determined by limb motion, foot anatomy and substrate consistency, thus the studies of fossil tracks can provide insight into producer, behaviour and palaeoenvironment, representing a direct window into the lives of extinct organisms (Falkingham, 2014). Triassic tetrapod tracks have a wide distribution across the supercontinent Pangea (see Lucas, 2007; Klein & Lucas, 2010a; and references herein). The Triassic track record is archosaur, lepidosauromorph/archosauromorph-(Rhynchosauroides) and synapsid-dominated (Haubold, 1971; Haubold, 1984; Klein & Haubold, 2007), and includes the oldest known dinosaur tracks (Klein & Lucas, 2010a). Several recent papers have asserted the usefulness of Triassic ichnotaxa for establishing correlations between different stratigraphic units on a global scale, with emphasis on the German and North American records (Lucas, 2007; Klein & Haubold, 2007; Klein & Lucas, 2010a). Footprints are the only tetrapod fossils known in many places, thus they provide important data on vertebrate distribution in space and time (Lucas, 2007). For instance, the number and diversity of chirotheriid tracks are such that several ichnologists have seriously proposed that it is easier to study the evolution of Triassic archosaurs through their abundant tracks than through their sparse skeletal remains (Lockley & Meyer, 2000). The Triassic archosaur tracks show a distinct stratigraphic distribution pattern (limited temporal ranges) that can be ascribed to different evolutionary developments of the locomotor apparatus (Klein & Lucas, 2010a). Nevertheless, Klein & Lucas (2010a) have suggested that the “single largest problem with Triassic footprint biostratigraphy and biochronology is the nonuniform ichnotaxonomy and evaluation of footprints that show extreme variation in shape due to extramorphological (substrate-related) phenomena”. Recent studies on dinosaur tracks have shown how the substrate can influence the final track shape with significant variations within the same trackway (e.g., Razzolini et al., 2014). For instance in a Triassic context, the ichnogenus Chirotherium Kaup, 1935a, is one of the described ichnotaxa with the most ichnospecies, but in several recent papers some of the ichnospecies described have been considered to be extramorphological variations or synonyms of well-established ichnotaxa (Klein & Haubold, 2007; Klein & Lucas, 2010a; Xing et al., 2013).

In the Iberian Peninsula the Triassic track record has gone almost unnoticed because of its scarcity and the fact that many of the tracks were described more than a century ago (e.g., Calderón, 1897; Navás, 1904; Navás, 1906; Gómez de Llarena, 1917). In the last few years new discoveries and reviews of previous material have notably increased what is known of the Iberian Triassic tetrapod track record (Gand et al., 2010; Díaz-Martínez & Pérez-García, 2011; Díaz-Martínez & Pérez-García, 2012; Fortuny et al., 2011). The latter authors made an exhaustive review of the Triassic bone and track record in the Iberian Peninsula, putting special emphasis on the paleobiogeography. Taking into account these recent papers, 26 localities with Triassic vertebrate tracks have been described since 1897 in the Iberian Peninsula (see Díaz-Martínez & Pérez-García, 2011; Díaz-Martínez & Pérez-García, 2012; Fortuny et al., 2012; Meléndez & Moratalla, 2014). Most of the studies predate the 1990s, and almost all the Iberian tracks have been studied just once and only took into account their ichnotaxonomic affinities. There are some examples where the material has been reassessed, such as Chirotherium catalaunicum Casanovas Cladellas, Santafé Llopis & Gómez Alba, 1979 (Fortuny et al., 2011), the Chirotherium tracks from Mallorca (Calafat et al., 1987; Gand et al., 2010), Chirotherium barthii Kaup, 1935b from Catalonia (Calzada, 1987; Valdiserri, Fortuny & Galobart, 2009), and the “Rillo de Gallo footprint” in Guadalajara (Calderón, 1897; Díaz-Martínez & Pérez-García, 2012). These reassessments have changed the initial identifications, and the age of the track-bearing layers has been taken into consideration. A number of researchers (Gand et al., 2010; Fortuny et al., 2011; Díaz-Martínez & Pérez-García, 2012) have emphasized the need to reappraise the Iberian Triassic vertebrate record in order to compare it with that from other coeval basins.

In the present work, we reassess the two slabs from the Moncayo massif (NE Spain) where Chirotherium ibericus (Navás, 1906) was defined (Navás, 1904; Navás, 1906). Since its definition, no one has yet reanalyzed this material firsthand, although it has been addressed in some ichnotaxonomic discussions (Leonardi, 1959; Kuhn, 1963; Haubold, 1971). During visits to the Natural Science Museum of the University of Zaragoza (Zaragoza, Spain), we have identified in the slabs new vertebrate tracks and anatomical details undescribed by Navás (1904), Navás (1906) and Leonardi (1959). Moreover, on the basis of recent geological studies (e.g., Díez et al., 2007; Bourquin et al., 2007; Bourquin et al., 2011), we are able to refine the geological location of these slabs (Navás site from here). The main aim of this paper is to discuss the ichnotaxonomy of all the vertebrate tracks found in the two slabs (those classified as Chirotherium ibericus and the other new material associated with them). Furthermore, we review the main tetrapod track assemblages of the Iberian Triassic (only including those localities that are well dated) in order to compare them with the biochrons based on tetrapod footprints (e.g., Klein & Haubold, 2007; Klein & Lucas, 2010a) proposed for the Triassic.

History of Chirotherium Ibericus

The Chirotherium ibericus tracks were found in the summer of 1895 when the Jesuit naturalist Longinos Navás was on a field trip in the Moncayo area. The summer visitor Mr. Ignacio de Inza showed Navás the place where “two dog-like traces” were imprinted cloven on the rock. Navás (1904) and Navás (1906) went on to identify six fossil tracks in this outcrop. His publications on Triassic tracks (Navás, 1904; Navás, 1906) reported the first occurrence of vertebrate tracks in Spain following the discovery of a chirotheriid footprint in the Triassic of Molina de Aragón, Guadalajara province (Calderón, 1897; Díaz-Martínez & Pérez-García, 2012). The first report of the discovery was in 1904, when Navás (1904) cited the presence of “Cheirotherium” in the Moncayo massif, including a first drawing of the slab bearing six ichnites made in the field by himself (Fig. 1). Subsequently, Navás (1906) assigned the tracks to a new ichnotaxon, Chirosaurus ibericus, but without a distinctive diagnosis. Nevertheless, it cannot be considered a nomen nudum because he provided a detailed description and compared it with other ichnotaxa (see art. 10.1 International Commission on Zoological Nomenclature, 1999). At the end of Navás’s (1906) paper, he proposed the possibility of using the name Chirotherium ibericum instead of Chirosaurus ibericus. In this case, Chirosaurus ibericus has priority over Chirotherium ibericum, which is a junior synonym, since the former was used before the latter. On the other hand, the ichnogenus Chirotherium has priority with respect to Chirosaurus (see Sarjeant, 1990) so the correct way to name the ichnotaxon proposed by Navás is Chirotherium ibericus.

Figure 1 Reproduction of the original drawing of Triassic ichnites.

Reproduction of the original drawing of slab CS.DA.39 bearing Triassic ichnites from the Moncayo massif, made by Longinos Navás in 1895 in the field and reported by Navás (1904, p. 149).

Navás (1906) proposed these tracks as a new ichnotaxon mainly on the basis of their age, size and shape. He suggested a Silurian age for the tracks, but all the other known Chirotherium tracks are Triassic. In addition, he compared the size of these tracks with the tracks from Molina de Aragon (Guadalajara, Spain) and those from the “British Museum of London” (today the Natural History Museum of London), concluding that the latter were much larger. He also suggested that the digit impressions of C. ibericus were more slender than the other tracks to which he compared them.

The slab was excised and new tracks appeared inside that were only cited but not described by Navás (1906). Finally, Navás (1906) proposed an amphibian as the trackmaker.

Subsequently, Leonardi (1959) re-studied the material of Navás (1906) on the basis of the previous publications and assigned the tracks from one slab to Chirotherium ibericus and the tracks from the other slab to Chirotherium coltoni (=Isochirotherium coltoni) Peabody (1957). Leonardi (1959) proposed that the presence of Chirotherium indicated a Triassic age. Finally, Kuhn (1963) and Haubold (1971) analysed the entire bibliography on pre-Cenozoic amphibian and reptile tracks and considered the tracks of the Navás site to be Chirotherium ibericum and Chirotheriidae indet., respectively.

Geological Setting

The tracks studied here are located in two excised slabs of fine-grained, bluish gray sandstones. According to the known data (Navás, 1906; Leonardi, 1959; Bastero Monserrat, 1989), the Navás site was located in a block of rock within Holocene deposits from the Moncayo massif, in the western part of Zaragoza province, NE Spain. The exact location is beside the road to the Moncayo Sanctuary, 700 m before the sanctuary (Fig. 2). The Navás site is located in the Aragonese Branch of the Iberian Range (Fig. 2). The Triassic of this region is composed of typical Germanic facies: detritic Buntsandstein, dolomitic Muschelkalk and lutitic-evaporitic Keuper (Arribas, 1985). The Moncayo massif is a structural relief that stands out from the surrounding topography and has a great richness of glacial and periglacial landforms (e.g., Pellicer & Echeverría, 2004). These Holocene deposits (e.g., block slopes) are formed from reworked material from the outcropping Buntsandstein facies of the Moncayo anticline (Fig. 2, Ramírez del Pozo, 1980).

Figure 2 Geological setting of the Triassic outcrops in the Moncayo Massif.

(A) Simplified geological map of the Iberian Peninsula. (B) General map of the Triassic outcrops and pictures from the Navás site. Map redrawn from MAGNA (Ramírez del Pozo, 1980).

The local series in the Moncayo outcrops is formed from Permo-Triassic detritic deposits lying unconformably on a Variscan basement (Arribas, 1985; Díez et al., 2007). This detritic series, lithologically composed of conglomerates, sandstones and lutites, is divided into four units: the Araviana, Tierga, Calcena and Trasobares units, in ascending stratigraphic order (Arribas, 1985). The basal conglomerates and lutites of the Araviana unit are attributed to the Permian based on paleopalynological data, whereas above them a noticeable hiatus has been recognized for the Lower Triassic (Díez et al., 2007). The Buntsandstein facies sensu stricto is represented by the Tierga, Calcena and Trasobares units, which are Anisian (Middle Triassic) in age based on paleopalynological data and sequence stratigraphy (Díez et al., 2007; Bourquin et al., 2007; Bourquin et al., 2011).

The studied track-bearing slabs were recovered within Holocene deposits from the NE slope of the Moncayo peak (Fig. 2); their exact stratigraphic origin cannot be specified with certainty. However, the lithological features and the nearest outcrops allow us to assign these slabs to Anisian Buntsandstein s.s. deposits, it being impossible to pinpoint their provenance specifically to one of the three local units. These deposits constitute a major cycle that can be divided into two minor cycles (Díez et al., 2007). The sandy nature of the slabs suggests that they probably belong to the Tierga-Calcena cycle in its retrogradational phase (mainly the Tierga unit), which is attributed to the lower Anisian (Díez et al., 2007). The Tierga unit—about 250 m thick and mainly composed of fine to medium-grained sandstones, with interbedded silty claystones—shows an evolution from a braided river to a fluvio-lacustrine environment, whereas the overlying Calcena unit—far less thick and rich in lutite—represents heterolithic coastal plain deposits (Díez et al., 2007).

Buntsandstein facies in the Iberian Range have traditionally been considered to be fluvial in origin (e.g., Arché & López-Gómez, 2006). Nonetheless, it should be noted that recently the red Buntsandstein sandstones of the south-eastern Aragonian Branch of the Iberian Chain have been reported as an evolving erg system (Soria et al., 2011), in accordance with the highly arid conditions predicted by paleoclimatic models for Western Europe during the Early Triassic (Péron et al., 2005).

Material and Methods

The analysed materials are two slabs, CS.DA.38 and CS.DA.39, which are housed in the Museo de Ciencias Naturales de la Universidad de Zaragoza, Zaragoza, Spain. The slabs have been deposited in the current institution since the late 20th century and were previously part of the collection of the Jesuit school of Zaragoza (Colegio El Salvador) at which Longinos Navás was teaching. The tracks were drawn using a large sheet of plastic. All the tracks were photographed individually, were measured (Fig. 3) and were labeled with the acronyms CS.DA.38.X or CS.DA.39.X (Figs. 4–6), depending on the slab and the position within the slab. CS.DA is the official label assigned by the Jesuit school and later maintained in the Natural Science Museum of the University of Zaragoza. In addition, m/p refers to manus (forelimb) and pes (hindlimb) tracks respectively.

Figure 3 Scheme used for the measurements of the tracks and trackways.

Scheme and measurements based on Demathieu & Wright (1988) and Clark, Aspen & Corrance (2002) for: (A) chirotheriid tracks, (B) Rhynchosauroides tracks, (C) tridactyl tracks, (D) trackways. Abbreviations in ‘Material and Methods’.

Figure 4 Picture and sketch map of slab CS.DA.38.

Figure 5 Picture and sketch map of slab CS.DA.39.

Figure 6 Sketch map of slabs CS.DA.38 and CS.DA.39 with the acronyms of each track.

The slabs have dimensions of 1.3 m length by 0.88 m width and 0.14 m thickness. The tracks which Navás sketched and identified as a single trackway in the papers of 1904 and 1906 in slab CS.DA.39 (Navás, 1904) are in fact part of two incomplete trackways (CS.DA.39.1.1p, CS.DA.39.1.1m, CS.DA.39.1.2p, CS.DA.39.1.2m and CS.DA.39.2.1m and one isolated track CS.DA.39.9) (Figs. 1 and 4–6). The tracks in slab CS.DA.39 are at the bottom and are stratigraphically beneath slab CS.DA.38. The natural casts of CS.DA.38 are located on the top of CS.DA.39.

Within slab CS.DA.38 (Figs. 4 and 6) we have identified three partial trackways (CS.DA.38.1–CS.DA.38.2 and CS.DA.38.4), a manus-pes track set (CS.DA.38.3) and three isolated tracks (CS.DA.38.5–CS.DA.38.7). In slab CS.DA.39 (Figs. 5 and 6), three partial trackways (CS.DA.39.1–CS.DA.39.3), five tracks (CS.DA.39.4–CS.DA.39.8) that could represent a trackway, and two isolated tracks (CS.DA.39.9–CS.DA.39.10) have been studied. In total, 28 vertebrate tracks have been studied (12 in CS.DA.38 and 18 in CS.DA.39).

Measurements were taken mainly according to Demathieu & Wright (1988) and Clark, Aspen & Corrance (2002) (see Fig. 3). Ichnotaxonomic discussions are mainly based on Avanzini & Renesto (2002), Demathieu & Demathieu (2004), Fichter & Kunz (2004), King et al. (2005) and Valdiserri & Avanzini (2007). In analysing and describing the skin marks we follow Avanzini (2000) and Kim et al. (2010).

The measurements taken were (Fig. 3; Tables 1–3): L, track length; l, track width; M, length set of I–IV; m, width set I–IV; I, length digit I; II, length digit II; III, length digit III; IV, length digit IV; V, length digit V; t, divarication II–IV; t′, divarication I–IV; f, divarication I–V; PL, pace length; Apm, angle between pes and manus; and Dpm, distance between pes and manus. All parameters are given and compared in cm, except t, t′, f, and Apm, which are given in degrees.

Table 1 Measurements of the Chirotherium barthii tracks from the Navás site.

Measurements (in cm and degrees) of the Chirotherium barthii tracks from the Navás site. Abbreviations are listed in ‘Material and Methods’.

	L	M	l	m	I	II	III	IV	V	t	t′	f	PL	pl	Apm	Dpm	
38.1.1p	11.7	8	–	5.6	3.7	5.4	7.5	6.1	–	25	39	78	33.8	–	21	11.3	
38.1.1m	4.7	3.3	–	–	–	1.4	2.3	2.8	2.3	45	–	–	–	–	–	–	
38.1.2p	11.2	8	7.5	6.1	–	–	–	6.1	3.7	23	45	85	–	–	–	–	
38.2.1p	–	–	–	–	–	3.7	5.2	4.2	–	20	28	–	35	–	30	11.8	
38.2.1m	–	–	–	–	–	1.4	1.8	–	–	30	–	–	–	36	–	–	
38.2.2p	–	–	–	–	–	–	–	–	–	–	–	–	–	–	–	11.8*	
38.2.2m	–	–	–	–	–	–	–	–	–	–	–	–	–	–	–	–	
38.3.1p	–	–	–	–	–	–	–	–	–	–	–	–	–	–	20	11.8*	
38.3.1m	4.7	2.8	–	–	–	1.4	1.8	2.4	1.8	41	–	–	–	–	–	–	
39.1.1p	14.5	8.9	8.9	7.5	–	7.9	9.4	7.4	5.2	29	43	79	42	–	–	14.1	
39.1.1m	–	–	–	–	–	–	–	3.3	2.8	–	–	–	–	38.5	–	–	
39.1.2p	13.1	8.9	7.9	7.9	2.8	6.1	7.5	6.6	4.7	18	42	85	–	–	14	11.8	
39.1.2m	5.6	4.2	6.1	4.7	1.4	3.3	3.8	3.3	3.3	33	65	145	–	–	–	–	
39.2.1p	13.1	–	–	–	–	–	–	–	–	–	–	–	45.1	–	–	16.4	
39.2.1m	6.1	–	–	–	–	3.3	4.2	4.2	3.7	48	–	–	–	–	–	–	
39.2.2p	–	–	–	–	–	–	–	–	5.2	–	–	86	–	–	–	–	
Notes.

* Estimate.

Table 2 Measurements of the Rhynchosauroides tracks from the Navás site.

Measurements (in cm and degrees) of the Rhynchosauroides tracks from the Navás site. Abbreviations are listed in ‘Material and Methods’.

	L	l	m	I	II	III	IV	V	t	t′	f	
39.4	4.6	2.7	2.4	1.6	2	2.5	2.8	0.8	10	50	78	
39.5	–	–	–	0.9	1.7	2	2.6	–	15	30	–	
39.6	–	–	–	–	–	–	–	–	–	–	–	
39.7	–	–	–	–	–	–	–	–		–	–	
39.8	–	–	–	–	–	–	–	–	–	–	–	
39.10	4.6	–	–	–	1.3	1.7	2.3	2.2	13	–	–	

Table 3 Measurements of the undetermined tracks from the Navás site.

Measurements (in cm and degrees) of the undetermined tracks from the Navás site. Abbreviations are listed in ‘Material and Methods’.

	L	l	II	III	IV	t	PL	
38.4.1	2	2.8	1.7	1.8	1.5	48	37	
38.4.2	2.4	2.5	1.7	1.9	1.8	35	–	
38.5	2.3	1.6	1.4	1.9	1.4	18	–	
38.6	2.3	–	1.4	1.6	–	–	–	
38.7	2.4	–	1.7	1.7	–	–	–	
39.11	2	2.2	1.4	1.8	1.4	12	–	

Furthermore, the entire bibliography relating to the record of Iberian Triassic tracks is revised in order to allow comparison with the global tetrapod track biochronology proposed by Klein & Haubold (2007) and Klein & Lucas (2010a). The information that we used is presented in simplified form in Table 4 and in the Table S1.

Table 4 Summary of the published Triassic tracks from the Iberian Peninsula that are located in a concrete chronostratigraphic age.

Only the most recent ichnotaxonomic determination is considered.

Icnotaxon	Age	Reference	
Dicynodontipus isp.	Anisian	Valdiserri, Fortuny & Galobart (2009)	
Procolophonichnium isp.	Anisian	Valdiserri, Fortuny & Galobart (2009)	
Rhynchosauroides isp.	Anisian (Fortuny et al., 2012)	Calzada (1987)	
Rhynchosauroides cf. beasleyei	Anisian (Fortuny et al., 2012)	Calzada (1987)	
Rhynchosauroides isp.	Anisian	Valdiserri, Fortuny & Galobart (2009)	
Rhynchosauroides isp.	Olenekian-Anisian	Gand et al. (2010)	
Rhynchosauroides isp.	Anisian	Gand et al. (2010)	
Rhynchosauroides isp.	Anisian	In this work	
Rotodactylus sp.	Anisian	Valdiserri, Fortuny & Galobart (2009)	
Brachychirotherium cf. gallicum	Upper Triassic?	Pérez-López (1993)	
Brachychirotherium gallicum	Anisian	Gand et al. (2010)	
Brachychirotherium isp.	Anisian	Gand et al. (2010)	
Chirtotherium barthii	Anisian (in this work)	In this work	
Chirotheium barthii	Anisian (Fortuny et al., 2012)	Calzada (1987)	
Chirotherium barthii	Anisian	Valdiserri, Fortuny & Galobart (2009)	
Chirotherium barthii	Anisian	Gand et al. (2010)	
Chirotherium isp.	Anisian	Gand et al. (2010)	
Isochirotherium soergeli	Anisian	Valdiserri, Fortuny & Galobart (2009)	
Isochirotherium cf. coureli	Anisian	Gand et al. (2010)	
Synaptichnium isp.	Anisian (Fortuny et al., 2012)	Calzada (1987)	
Synaptichnium isp.	Anisian	Valdiserri, Fortuny & Galobart (2009)	
Chirotheriid	Ladinian-early Carnian	Fortuny et al. (2012)	
Chirotheriid	Ladinian	Meléndez & Moratalla (2014)	
Chirotheriid	Anisian	In this work	
Eubrontes isp.	Rhaetian	Pascual-Arribas & Latorre-Macarrón (2000)	
Anchisauripus isp.	Rhaetian	Pascual-Arribas & Latorre-Macarrón (2000)	
Coelurosaurichnus perriauxi	Anisian	Gand et al. (2010)	
Paratrisauropus latus	Anisian	Gand et al. (2010)	
Archosauria	Landian	Demathieu, Pérez-López & Pérez-Lorente (1999)	

Systematic Ichnology

Ichnofamily Chirotheriidae Abel, 1935	
Ichnogenus Chirotherium Kaup, 1835a	
Chirotherium barthii Kaup, 1835b	
(Figs. 4–8)	

1904 Cheirotherium Navás, p. 149.	
1906 Chirosaurus ibericus Navás, p. 208, Figs. 2 and 3.	
1906 Chirotherium ibericum Navás, p. 213, Figs. 2 and 3.	
1959 Chirotherium ibericus Leonardi, p. 243, photograph 3.	
1959 Chirotherium coltoni Leonardi, p. 243.	
1963 Chirotherium ibericum Kuhn, p. 71.	
1971 Chirotheriidae indet. Haubold, p. 58.	

Referred specimens: CS.DA.38.1.1p, CS.DA.38.1.1m, CS.DA.38.1.2p, CS.DA.38.2.1p, CS.DA.38.2.1m, CS.DA.38.2.2p, CS.DA.38.2.2m, CS.DA.38.3.1p, CS.DA.38.3.1m, CS.DA.39.1.1p, CS.DA.39.1.1m, CS.DA.39.1.2p, CS.DA.39.1.2m, CS.DA.39.2.1p, CS.DA.39.2.1m and CS.DA.39.2.2p.

Material: 16 tracks (four partial trackways and one pes/manus set) in the two slabs (nine in CS.DA.38 and seven in CS.DA.39); some of them show skin and phalangeal pad impressions (Figs. 4–8; Table 1).

Figure 7 Pictures of the studied tracks assigned to Chirotherium barthii.

(A) CS.DA.38.1.1p and CS.DA.38.1.1m. (B) CS.DA.38.1.2p. (C) CS.DA.39.1.1p. (D) CS.DA.39.1.2m (see location in Fig. 6).

Figure 8 Main Chirotherium ichnospecies compared with the Navás site tracks.

(A) C. vorbachi (redrawn from King et al., 2005). (B) C. sickleri (redrawn from Haubold, 1971). (C) C. lulli (redrawn from Baird, 1954). (D) C. lomasi (redrawn from Baird, 1957). (E) C. storetonense (redrawn from King et al., 2005. (F) C. rex (redrawn from Peabody, 1957. (G) C. wondrai (redrawn from Haubold, 1971. (H) C. coureli (redrawn from Demathieu, 1970). (I) C. barthii (redrawn from Haubold, 1971). J: CS.DA.38.1.1p. K: CS.DA.38.1.2p. and L: CS.DA.39.1.1p.

Horizon and locality: Buntsandstein facies, Anisian (Middle Triassic); Navás site (Moncayo massif, Zaragoza, Spain).

Description:

Manus: There are seven manus tracks but only one is complete, CS.DA.39.1.2m. It is pentadactyl, mesaxonic, asymmetric and digitigrade (Fig. 7). The length of the manus tracks varies from 4.7 cm to 6.1 cm, and the width of the only complete track is 6.1 cm. Four digit impressions (I–IV) are directed forward, and one, the digit V impression, is directed laterally. Digit I is often poorly preserved or absent. There is little difference in the length of digits III and IV, which are longer than digits I (the smallest) and II. Digit V is situated proximally below digit IV. It is divergent (from the long axis through digit III) and separated from the other digits. Digits I, II, III and IV fuse at their proximal ends but do not present clear metacarpal pads. At least four of the digits (I–IV) have an acuminate end, although these are not as prominent as those on the pes. The divarication angle II–IV is from 30° to 48°. The angulation between digits I–IV and I–V is 65°and 145°respectively in CS.DA.39.1.2m (see Table 1).

The manus tracks are more poorly-preserved than the pes tracks. The manus is relatively small compared to the pes, with the manus-pes length ratio ranging from 0.4 to 0.46.

Pes: These are pentadactyl, mesaxonic, asymmetric and semiplantigrade tracks (see Fig. 7). Four digit impressions (I–IV) are directed forward, and one, the digit V impression, is directed laterally. They are longer than wide. The length of the pes print varies from 11.2 cm to 14.5 cm, and the width ranges from 7.5 cm to 8.9 cm. The length to width ratio varies from 1.5 to 1.65. Digits I–IV form an isolated group that is longer (from 8 to 8.9 cm) than wide (from 5.6 to 7.9 cm).The digits are longer than wide and have an acuminate end. Digit III is slightly longer than digit IV and digit II. Digit I is the smallest (III>IV>II>I); it is located posteriorly and is usually the worst preserved. The divarication angle II–IV varies from 18° to 29°and I–IV from 28° to 45°. Digits I–IV show clear impressions of digital pads, but not metatarsal pads. Digit V is rotated outwards with respect to digit IV. It shows a subovoid impression of the metatarsal pad. The angulation between digit I–V varies from 78° to 86°. In the pes track CS.DA.38.1.2p skin impressions are recognizable. Another part of the slab with skin-like marks has been found, but there are not any tracks associated with it. In both cases, they are very small in size, about 1 mm on the digit V surface (Fig. 9A). Their shape is predominantly subrounded and does not show a distinct ornamentation. Impressions are separated by a thin and non-imbricated depression.

Figure 9 Photographs of the new identified material.

(A) Isolated set of skin impressions from the slab CS.DA.38 (see location in Fig. 6). (B) Chirotheriidae indet., CS.DA.39.3.2p. (C) Undetermined material (unnamed morphotype), CS.DA.38.4.1. (D) Rhynchosauroides isp., CS.DA.39.4. (E) Rhynchosauroides isp., CS.DA.39.9. (F) Rhynchosauroides isp., CS.DA.39.5.

Trackway: There are four partial trackways and one manus-pes set (see Figs. 4–7). The manus is rotated outward 14°–30° with respect to the pes. The manus/pes distances range from 11.3 cm to 16.4 cm. The manus is placed in front of, and to the inside of, the pes (usually with the outer edge of the manus in line with the outer edge of the pes). The pace length between pes tracks is from 33.8 cm to 42 cm, and between manus tracks from 36 cm to 38.5 cm.

Remarks:

The tracks in both slabs have the same general shape. Although there is slight variability among them, we consider that this variability is a consequence of preservational factors. The main difference between the tracks is the size. The tracks in CS.DA.38 are slightly smaller than the CS.DA.39 tracks (see Table 1). Nevertheless, we consider that size is not a valid ichnotaxobase (see Bertling et al., 2006), and therefore we have classified all of them in the same way.

Since the pes tracks are semiplantigrade and pentadactyl with a compact anterior digit I–IV group and a posterolaterally positioned digit V, and the manus tracks are smaller than the pes tracks, pentadactyl, mesaxonic, asymmetric and digitigrade, they can be attributed to the ichnofamily Chirotheriidae (cf. Demathieu & Demathieu, 2004; King et al., 2005).

Demathieu & Demathieu (2004) and King et al. (2005) proposed the proportions of pes digits I–IV as the most important feature for distinguishing chirotheriid ichnogenera, whereas the length, shape and position of digit V are variable (Klein & Haubold, 2003). The ichnofamily Chirotheriidae is composed of nine ichnogenera: Brachychirotherium Beurlen, 1950; Chirotherium; Isochirotherium Haubold, 1971; Paleochirotherium Fichter & Kunz, 2011; Parachirotherium Kuhn, 1958; Protochirotherium Fichter & Kunz, 2004; Parasynaptichnium Mietto, 1987; Sphingopus Demathieu, 1966; and Synaptichnium Nopcsa, 1923. Five of these, Brachychirotherium, Chirotherium, Isochirotherium, Parachirotherium and Sphingopus, are mesaxonic, and only in three of these, Brachychirotherium, Chirotherium and Isochirotherium do the digit I–IV impressions form an isolated group. The tracks from the Navás site belong to Chirotherium because the digit IV impression is normally longer than II and the digit II–IV impressions are similar in thickness. In Isochirotherium digit II is always longer than digit IV, and in Brachychirotherium digits II and III are thicker than digits I, IV and V (sensu Haubold, 1971; King et al., 2005).

The studied material, classified as C. ibericus by Navás (1906), and other material of the same shape, presents the digit III impression slightly longer than digits II and IV. This character differentiates it from C. vorbachi Kirchner, 1927 (Fig. 8A), which is much more mesaxonic. Furthermore, it is characterized by a digit IV impression that is slightly longer and often thinner than digit II. It differs from C. sickleri Kaup, 1835c, C. lulli Bock, 1952, and C. eyermani Baird, 1957, which present digit IV clearly longer than digit II (Figs. 8B–8D), and from C. storetonense Morton, 1863, which has digit II thinner than digit IV (Fig. 8E). Additionally, the digit I impression is smaller and thinner than the digit II–IV impressions, and located forwardly and slightly independently with respect to digits II–IV. These characters differentiate it from C. rex Peabody, 1948, C. wondrai Heller, 1952, and C. coureli Demathieu, 1970, which have a more robust digit I impression positioned at the same proximal position as the other digits and forming a more compact group I–IV (Figs. 8F–8H). The only ichnotaxon that shares all the above-described characters with the studied material is C. barthii (Fig. 8I). Only size differentiates them from one another. The Navás site tracks (Figs. 8J–8L) are smaller than the holotype of C. barthii. Nevertheless, we consider that size is not a valid ichnotaxobase because it can merely represent an ontogenetic variation. Accordingly, we regard the two types of track as the same. C. barthii was defined in 1835 by Kaup on the basis of Middle Triassic tracks from Germany. Therefore, C. barthii has temporal priority with respect to the ichnotaxon C. ibericus, and the latter is a junior synonym of C. barthii.

Ichnofamily Chirotheriidae Abel, 1935	
Chirotheriidae indet.	
(Figs. 4–6 and 9)	

Referred specimens: CS.DA.39.3.1 and CS.DA.39.3.2.

Material: A possible partial trackway of pes tracks in slab CS.DA.39 (Figs. 4–6 and 9B; Table 2).

Horizon and locality: Buntsandstein facies, Anisian (Middle Triassic); Navás site (Moncayo massif, Zaragoza, Spain).

Description:

The tracks are poorly-preserved and could be two consecutive pes tracks. The first track is pentadactyl, mesaxonic, asymmetric and semiplantigrade (Fig. 9B). Four digit impressions (I–IV) are directed forward, and one, the digit V impression, is directed laterally. It is longer than wide. The second track preserves the digit V impression, which is also directed laterally, and some impressions directed forwards, which could belong to any of the digit I–IV impressions. The pace length is 72 cm.

Remarks:

As pointed out in the previous section, pes tracks that are semiplantigrade and pentadactyl with a compact anterior digit I–IV group are related with the ichnofamily Chirotheriidae (cf. Demathieu & Demathieu, 2004; King et al., 2005). Nevertheless, we are not assigning these tracks to a concrete chirotheriid ichnogenus because the proportions of digits I–IV are the most important feature for classification (Demathieu & Demathieu, 2004; King et al., 2005) and this information cannot be extracted from the tracks due to their state of preservation.

Ichnofamily Rhynchosauroidae Haubold, 1966	
Ichnogenus Rhynchosauroides Maidwell, 1911	
Rhynchosauroides isp.	
(Figs. 4–6 and 9D–9F)	

Referred specimens: CS.DA.39.4, CS.DA.39.5, CS.DA.39.6, CS.DA.39.7, CS.DA.39.8 and CS.DA.39.9.

Material: Part of a possible trackway (CS.DA.39.4, CS.DA.39.5, CS.DA.39.6, CS.DA.39.7 and CS.DA.39.8) and an isolated track (CS.DA.39.9) in slab CS.DA.39 (Figs. 4–6 and 9A–9C; Table 2).

Horizon and locality: Buntsandstein facies, Anisian (Middle Triassic); Navás site (Moncayo massif, Zaragoza, Spain).

Description:

Manus: the best-preserved manus track, CS.DA.39.4 (Fig. 9D), is pentadactyl, ectaxonic, very asymmetric and plantigrade. Four digit impressions (I–IV) are directed forward, and one, the digit V impression, is directed more laterally. The length of the track is 3.7 cm and the width 2.4 cm (length/width ratio 1.54). The digits are longer than wide and rotated medially. Digit IV is the longest. Digit IV>III>II>I>V. The divarication angle II–IV is 10°, I–IV is 50° and I–V is 78°. The digit impressions show clear impressions of claw marks. The palm impression is well-marked and bilobed. Similar to this track is CS.DA.39.9 4 (Fig. 9E), but one of the digit impressions (probably the digit IV impression) is not preserved.

Pes: track CS.DA.39.5 4 (Fig. 9F) is tetradactyl, very asymmetric and digitigrade. The four digit impressions (I–IV) are longer than wide, directed forward and rotated medially. It is not possible to measure the length or width of the track due to its state of preservation. Digit IV is the longest. Digit IV>III>II>I>V. The divarication angle II–IV is 15° and I–IV is 30°. The digit impressions do not show clear impressions of claw marks.

Tracks CS.DA.39.6, CS.DA.39.7 and CS.DA.39.8 are tridactyl and didactyl. The shape and size of the preserved digit impressions are similar to those of tracks CS.DA.39.4 and CS.DA.39.5, and they are located close to them.

Remarks:

There is clear variability among all the tracks. Some of them, CS.DA.39.4–CS.DA.39.8, could be part of the same trackway given their shape, size and location. Therefore, this variability is probably a consequence of the state of preservation and not because they are different morphotypes. The best-preserved tracks present the following main features: four digit impressions (I–IV) directed forward; digits longer than wide and rotated medially; and digits increasing in length from I to IV. In addition, in CS.DA.39.4 and CS.DA.39.10 (manus tracks) there is a digit V impression, which is shorter than the others and is turned outwards. These characters are typical of the ichnogenus Rhynchosauroides (Melchor & de Valais, 2006; Hunt & Lucas, 2007a; Avanzini, Piñuela & García-Ramos, 2010; Lucas et al., 2010). However, more than 20 ichnospecies of Rhynchosauroides have been defined (see Haubold, 1971), and the validity of some of them has not been discussed. As we have suggested above, moreover, the shape of the tracks studied here is variable, and they are not well enough preserved for a confident determination of the ichnospecies. Accordingly, we have decided to be cautious in assigning these tracks to Rhynchosauroides isp.

Undetermined material	
Unnamed Morphotype	
(Figs. 4–6 and 9C)	

Referred specimens: CS.DA.38.4, CS.DA.38.5, CS.DA.38.6, CS.DA.38.7 and CS.DA.39.10.

Material: six footprints in the two slabs (five in CS.DA.38 and one in CS.DA.39); two of them are a pair 4 (Figs. 4–6 and 9C; Table 3).

Horizon and locality: Buntsandstein facies, Anisian (Middle Triassic); Navás site (Moncayo massif, Zaragoza, Spain).

Description:

These are tridactyl, mesaxonic, symmetric and digitigrade tracks. The length is from 2 cm to 2.4 cm, and the width from 1.6 cm to 2.8 cm. The three digit impressions are directed forward. There is little difference in the length of the digits, the central one being the longest. The divergence between the lateral digits is variable. The tracks of the pair CS.DA.38 (Figs. 4–6 and 9C) present a greater divarication angle than the other tracks. The digit impressions of these tracks are the thinnest as well. At least three tracks (CS.DA.38.4.1, CS.DA.38.4.2 and CS.DA.38.5) have an acuminate end.

The pace length in the pair CS.DA.38.4 is 37 cm.

Remarks:

Although some tracks are thinner than others, all the tracks present the same features. Tridactyl, mesaxonic and digitigrade tracks could be associated with non-avian or avian theropod tracks (cf. Thulborn, 1990; De Valais & Melchor, 2008). However, non-avian theropod tracks are generally asymmetric, and there are no avian remains in the Anisian. The tracks are very shallow and are not well-preserved. Their preservation is not easy to interpret. Thus, it may have been preserved as undertracks and/or they are in fact parts of other more complex kinds of track superimposed (e.g., chirotheriid and/or Rhynchosauroides). Because of the poor state of preservation of the specimens, any attribution would be tentative.

Discussion

The Navás site tracks and the Triassic Iberian record

After a reassessment of the Navás site, Chirotherium barthii, Chirotheriidae indet., Rhynchosauroides isp., and an unnamed morphotype have been identified. As at the Navás site, chirotheriid tracks are well-represented in other Iberian localities. This kind of tracks is the most abundant compared to other ichnogroups. According to the revision of Díaz-Martínez & Pérez-García (2011) and the most recent articles (Díaz-Martínez & Pérez-García, 2012; Fortuny et al., 2012; Meléndez & Moratalla, 2014; this work) on 63 classified remains in 26 publications, 26 correspond to chirotheriid tracks. These tracks have been attributed to Brachychirotherium (2), Chirotherium (13), Isochirotherium (3), Synaptichnium (5) and indeterminate chirotheriids (3). The re-evaluation of the type material of C. ibericus has demonstrated that it is a junior synonym of C. barthii. This latter ichnospecies has also been found at other Iberian localities such as Corral d’en Parera (Calzada, 1987) and in the Eslida Formation (Gand et al., 2010), both Anisian in age. Gand et al. (2010) suggested that the presence of C. barthii is “rather uncommon in Spain”. What is remarkable is the small size of the Iberian tracks assigned to C. barthii (Figs. 7A–7D), since in the emended description of the diagnosis of this ichnospecies provided by King et al. (2005), the authors proposed that C. barthii has a pes length of about 19–22 cm. In the case of the Iberian tracks, the tracks from the Navás site have a pes length of between 11 and 14 cm, whereas the tracks described by Gand et al. (2010) are even smaller (pes length 8.4 cm). Calzada (1987) did not measure the total length of the tracks but the length of digit III (9.5–9.6 cm) according to the scale of the track pictures also seems small in size. Small-sized C. barthii tracks have also been described in the Middle Triassic of the United States (Klein & Lucas, 2010b; Lovelace & Lovelace, 2012), Morocco (Tourani et al., 2010; Klein et al., 2011), and China (Xing et al., 2013), and possibly also Switzerland (Cavin et al., 2012). The small size of the Iberian tracks assigned to C. barthii would fit better with the pes length of C. sickleri. In fact, King et al. (2005) proposed that “there is a strong possibility that C. sickleri may represent the tracks of a juvenile reptile, whose adult tracks might be attributed to C. barthii or C. storetonense Morton, 1863”. Klein & Haubold (2003) also showed the similarities between the two ichnotaxa with a landmark analysis and suggested that “one could suspect a juvenile C. barthii”. The authors pointed out that some features of C. sickleri, such as the manus print morphology and the trackway pattern, were not included in the analysis, which was mainly done with the pes morphology. The Navás site, as well as the recent publications of small-sized C. barthii tracks, thus adds valuable data to this debate, and an exhaustive comparison of the two ichnotaxa is needed in order to discern whether C. sickleri is an ontogenetic variation of C. barthii or in fact a different ichnospecies.

The C. barthii pes track CS.DA.38.1.2p has preserved skin traces (Fig. 4) that are not noted in previous reports on the material. Other skin traces were found in the same slab (Fig. 9F), but they are not related with any visible track. The skin impressions were only created because the integument registered on a receptive substrate (Gatesy, 2001; Pérez-Lorente, 2001), and the motion of the skin relative to the sediment during separation strongly influences the morphology of the skin impression (Gatesy, 2001; Avanzini, Piñuela & García-Ramos, 2011). In this case, the ornamentation reveals scales that are sub-rounded to polygonal in shape, and it is present in digit V. These scale marks are similar to other chirotheriid skin impressions studied by Avanzini (2000), suggesting that these kinds of scales are similar to those of birds and extant Archosauria.

Six tracks belonging to Rhynchosauroides, including pes and manus tracks, were found at the Navás site. Rhynchosauroides is the best-known ichnogenus in the Triassic record of Iberia. It has been found at 13 localities in the provinces of Barcelona, Cantabria, Castellón, Guadalajara, Teruel and Zaragoza (Demathieu & Saiz de Omeñaca, 1976; Demathieu & Saiz de Omeñaca, 1977; Demathieu, Ramos & Sopeña, 1978; Demathieu & Saiz de Omeñaca, 1979; Calzada, 1987; Demathieu & Saiz de Omeñaca, 1990; Ezquerra et al., 1995; Valdiserri, Fortuny & Galobart, 2009; Gand et al., 2010; this work). Four Rhynchosauroides ichnospecies have been described in the Iberian Peninsula: Rhynchosauroides santanderensis Demathieu & Saiz de Omeñaca, 1976; Rhynchosauroides virgiliae Demathieu, Ramos & Sopeña, 1978; Rhynchosauroides extraneus Demathieu & Saiz de Omeñaca, 1979; and Rhynchosauroides simulans Demathieu & Saiz de Omeñaca, 1979. The temporal record of this ichnotaxon is predominantly Anisian, as exemplified by the Navás site, although it has also been described in the Permian (Valentini, Conti & Mariotti, 2007) and even in the Late Jurassic (Avanzini, Piñuela & García-Ramos, 2010).

Finally, undetermined material has also been found at the Navás site. These tracks are tridactyl and mesaxonic, but they are probably the preserved part of other tracks. In the Iberian record other Triassic tracks with problematic affinities have been cited (see Table S1). The tracks classified as type 3 and type 4 of Demathieu & Saiz de Omeñaca (1976) and Demathieu & Saiz de Omeñaca, 1977) are similar to those from the Navás site. In the former case, the shape of the tracks suggests that they are part of Rhynchosauroides tracks. It is therefore possible that the Navás tracks might be as well.

The Navás tracksite presents the Chirotherium barthii-Rhynchosauroides ichnoassemblage. This ichnoassemblage is common in other Middle Triassic localities in Iberia (Calzada, 1987; Gand et al., 2010), as well as in other ichnoassemblages with greater ichnodiversity described in the Middle Triassic of Europe (e.g., France, Gand, Demathieu & Montenat, 2007; Italy, Avanzini, Bernardi & Nicosia, 2011; Poland, Niedźwiedzki et al., 2007), North Africa (e.g., Morocco, Tourani et al., 2010; Klein et al., 2011) and North America (e.g., Hunt et al., 1993; Heckert, Lucas & Hunt, 2005). Analysis of the ichnoassemblage from the Navás site within the context of the global tetrapod track biochronology of the Triassic shows it to belong to biochron II (sensu Klein & Haubold, 2007) or the Chirotherium barthii biochron (sensu Klein & Lucas, 2010a). Both biochrons are defined for the upper Olenekian-lower Anisian age, which is coherent with the age of the Navás site, which is here considered lower Anisian.

In the case of the track record here described, the ichnogenera have been atributed to trackmakers belonging to different taxonomic categories in previous literature. The inferred trackmakers are Archosauriformes for Chirotherium as well as Lepidosauromorpha/Eosuchia for Rhynchosauroides (Klein et al., 2011; Avanzini, Piñuela & García-Ramos, 2011).

The Triassic record of vertebrate tracks in the Iberian Peninsula and the tetrapod-track-based biochrons

Several characteristic track assemblages and ichnotaxa have a restricted stratigraphic range and can therefore be repeatedly observed in the global record in distinct time intervals (Klein & Lucas, 2010a). Several authors (e.g., Haubold, 1969; Demathieu & Haubold, 1974; Olsen, 1980; Lockley & Hunt, 1995; Hunt & Lucas, 2007b; Lucas, 2007; Klein & Haubold, 2007; Klein & Lucas, 2010a; Xing et al., 2014; and references therein) have proposed the possibility of a tetrapod ichnostratigraphy of Triassic sequences. Nevertheless, vertebrate track biochronology faces three main problems that result in it being not as refined as tetrapod body fossils can be: the ichnotaxonomy, the evolutionary turnover rates and facies restrictions (Lucas, 2007). The last two biases are conditioned by the habitat and rate of evolution that is proper to each taxon and animal group (see discussion in Lucas, 2007). Thus the main problem with Triassic footprint biostratigraphy and biochronology is the nonuniform ichnotaxonomy and the evaluation of footprints that show extreme variation in shape due to extramorphological (substrate-related) phenomena (Klein & Lucas, 2010a). For instance, 75 chirotherian ichnospecies have been described from Triassic deposits in Europe, North America, South America, northern and southern Africa, and China (Klein & Haubold, 2007; Klein & Lucas, 2010a), but most of them may be synonyms and/or extramorphological variations of perhaps 35 valid ichnotaxa (Xing et al., 2013).

Since 1897, when the first work on Triassic vertebrate tracks from the Iberian Peninsula was published, 25 scientific works on the topic have been published (see Díaz-Martínez & Pérez-García, 2011; Díaz-Martínez & Pérez-García, 2012; Fortuny et al., 2012; Meléndez & Moratalla, 2014) (Table S1). Vertebrate tracks have been reported from 26 sites, and six new ichnotaxa have been defined: Chirotherium ibericus, R. santanderensis, R. virgiliae, Chirotherium catalaunicum, R. extraneus and R. simulans. More than half of the papers on Triassic tracks were published before the 1990s, and almost none of the Iberian tracks have been re-studied. In all the papers that reassess previously studied tracks, the initial ichnotaxonomic identifications and the age of the track-bearing layers were subsequently modified (e.g., Leonardi, 1959; Gand et al., 2010; Fortuny et al., 2011; Díaz-Martínez & Pérez-García, 2012, this work). In addition to the nonuniform ichnotaxonomy, the Iberian record presents another problem when it comes to comparisons with the biostratigraphy and biochronology proposed for the Triassic tracks. This is the temporal geological context of the ichnological localities. In some papers the age of the tracksite is well defined in terms of chronostratigraphic ages such as Anisian, Ladinian or Rhaetian (e.g., Pascual-Arribas & Latorre-Macarrón, 2000; Gand et al., 2010; Fortuny et al., 2011). In other papers, however, authors have located the tracks within the classic Germanic facies (Buntsandstein, Muschelkalk and Keuper) (see Díaz-Martínez & Pérez-García, 2011; Table S1), which are not considered time intervals, as the development of the different rift systems in central and western Europe was not coeval, causing diachronous facies changes (López-Gómez, Arché & Pérez-López, 2002; and references therein). In this context, we have only compared the Iberian record that is located in a concrete chronostratigraphic age (Table 4; Fig. 10) with the tetrapod track biochronology of the Triassic proposed by Klein & Haubold (2007) and Klein & Lucas (2010a).

Figure 10 Stratigraphic distribution of tetrapod track ichnotaxa and form groups in the Triassic with the global biochrons compared with the Iberian record. The global biochrons are based on Klein & Haubold (2007) and Klein & Lucas (2010a).

The red lines represent the Iberian record based on Table 4. Abbreviations: Atr., Atreipus; Grall., Grallator; Coelurosau., Coelurosaurichnus; Dicy., Dicynodont tracks; Prot., Protochirotherium.

Lowest Triassic-upper Lower Triassic

Klein & Lucas (2010a) define the “dicynodont-tracks” biochron for the latest Changhsingian-Induan stratigraphic interval, during which earliest Triassic dicynodont tracks are characteristic. The authors suggest that this biochron is so far restricted to Gondwana.

For the late Induan-late Olenekian stratigraphic interval, Klein & Haubold (2007) propose biochron I, and Klein & Lucas (2010a) the Protochirotherium biochron. The typical ichnological assemblage of these biochrons is based on the ichnotaxa Protochirotherium (Synaptichnium), Rhynchosauroides and Procolophonichnium Nopcsa, 1923 (Klein & Lucas, 2010a).

In the Iberian Peninsula the only record of Triassic tracks for this interval is composed solely of Rhynchosauroides tracks considered to be Olenekian-Anisian in age (Gand et al., 2010). This is the oldest Triassic track record in the Iberian Peninsula. The ichnotaxon Rhynchosauroides has a broad temporal distribution. Klein & Lucas (2010a) consider this ichnotaxon to range throughout the Triassic (it is very common in the Late Triassic, Hunt & Lucas, 2007a), and Avanzini, Piñuela & García-Ramos (2010) even identified Rhynchosauroides tracks in the Upper Jurassic of Asturias (Spain). The appearance of this ichnotaxon in Iberia is thus coherent with the global distribution proposed by Klein & Lucas (2010a). Nevertheless, the record is very scarce and does not give concrete data on the biochron, which could be within the Olenekian-Anisian time range given the dominance of Rhynchosauroides in some footprint assemblages (Fig. 10).

Uppermost Lower Triassic-Middle Triassic

For this interval Klein & Haubold (2007) proposed three biochrons, and Klein & Lucas (2010a) two. For the late Olenekian-early Anisian, biochron II (Klein & Haubold, 2007) and the Chirotherium barthii biochron (Klein & Lucas, 2010a) were defined. The typical assemblage for this temporal interval is composed of C. barthii, C. sickleri, Isochirotherium, Synaptichnium (“Brachychirotherium”), Rotodactylus Peabody, 1948, Rhynchosauroides, Procolophonichnium, dicynodont tracks and Capitosauroides Haubold, 1970 (Klein & Lucas, 2010a).

Klein & Haubold (2007) proposed biochron III for the late Anisian-early Ladinian interval and biochron IV for the late Ladinian. Biochron III is composed of the ichnotaxa Sphingopus, Atreipus Olsen & Baird, 1986, Grallator Hitchcock, 1858, Rotodactylus, Isochirotherium and Synaptichnium (“Brachychirotherium”). Typical of biochron IV are Parachirotherium, Atreipus, Grallator, and Synaptichnium (“Brachychirotherium”). For almost the same temporal range as biochrons III and IV, Klein & Lucas (2010a) defined the Atreipus-Grallator biochron in the late Anisian-lowermost Carnian. The typical assemblage of this biochron comprises Atreipus, Grallator (“Coelurosaurichnus”), Synaptichnium (“Brachychirotherium”), Isochirotherium, Sphingopus, Parachirotherium, Rhynchosauroides and Procolophonichnium.

The Iberian record in the uppermost Lower Triassic-Middle Triassic time interval is abundant. As suggested above, the oldest remains are Olenekian-Anisian in age and are composed only of Rhynchosauroides tracks (Gand et al., 2010). Calzada (1987) proposed a late Olenekian or early Anisian age for the tracks that he studied in the Buntsandstein of Catalonia, whereas Valdiserri, Fortuny & Galobart (2009) and Fortuny et al. (2012) suggested an Anisian age for these tracks based on magnetostratigraphy and biostratigraphic data. In the Anisian, the Iberian assemblage consists of Dicynodontipus Lilienstern, 1944, Procolophonichnium, Rhynchosauroides, Rotodactylus, Brachychirotherium, Chirotherium barthii, Isochirotherium, Synaptichnium, Coelurosaurichnus Huene, 1941, and Paratrisauropus Ellenberger, 1972 (Calzada, 1987; Valdiserri, Fortuny & Galobart, 2009; Gand et al., 2010; Fortuny et al., 2012; this work). In the Ladinian only three localities with vertebrate tracks have been described to date (Demathieu, Pérez-López & Pérez-Lorente, 1999; Fortuny et al., 2012; Meléndez & Moratalla, 2014). Demathieu, Pérez-López & Pérez-Lorente (1999) described tridactyl tracks and referred them to a crurotarsal/dinosauroid trackmaker. Fortuny et al. (2012) studied some vertebrate ichnites that were recovered from the Middle Muschelkalk (Ladinian-early Carnian) and classified them as belonging to the Chirotheriidae ichnofamily. Finally, Meléndez & Moratalla (2014) cited the presence of tracks with the general footprint morphology of the “group” formed by the Chirotherium–Isochirotherium–Brachychirotherium ichnogenera.

When the Iberian record for this temporal interval is compared with the tetrapod-track-based biochrons, it can be seen that several characteristic Triassic track assemblages and ichnotaxa with a restricted stratigraphic range are present. For instance, the ichnotaxon Chirotherium barthii has been found in four localities of an Anisian age (Table 4). The presence of this ichnotaxon is typical of biochron II of Klein & Haubold (2007) and the Chirotherium barthii biochron of Klein & Lucas (2010a), both from the late Olenekian-early Anisian interval. The latter authors suggest that Chirotherium barthii disappears during the Anisian. The ichnotaxa Isochirotherium and Rotodactylus have been found in the Anisian of the Iberian Peninsula as well. Both ichnotaxa have a broader distribution (late Olenekian-early Ladinian) than C. barthii, forming part of biochrons II and III of Klein & Haubold (2007) and the C. barthii and Atreipus-Grallator biochrons of Klein & Lucas (2010a). These ichnotaxa disappear before the end of the Ladinian (Klein & Haubold, 2007). Also present in the Anisian of the Iberian Peninsula are the ichnotaxa Coelurosaurichnus and Paratrisauropus. Coelurosaurichnus is present in biochron III (late Anisian-early Ladinian) of Klein & Haubold (2007) and in the Atreipus-Grallator biochron (late Anisian-lowermost Carnian) of Klein & Lucas (2010a). The ichnotaxon Synaptichnium, present in the Anisian of Iberia, is typical of biochrons II, III and IV of Klein & Haubold (2007) and the C. barthii and Atreipus-Grallator biochrons of Klein & Lucas (2010a) for the late Olenekian-Ladinian time range. The ichnotaxon Brachychirotherium was cited in the Anisian of the Iberian Peninsula by Gand et al. (2010). Nevertheless, Klein & Haubold (2007) and Klein & Lucas (2010a) placed this ichnotaxon in biochrons V and VI, and in the Brachychirotherium biochron of the lowermost Carnian to Rhaetian respectively. After analyzing the tracks classified as Brachychirotherium by Gand et al. (2010), we conclude that they present a Chirotherium affinity (the digit IV impression is longer than II, and the digit II–IV impressions are similar in thickness). In this case, the age of these tracks matches with the distribution of Chirotherium in the biochronological approaches. Other ichnotaxa with a broad temporal distribution (see Klein & Lucas, 2010a), such as Dicynodontipus, Procolophonichnium and Rhynchosauroides, have also been found in the Anisian of the Iberian Peninsula.

For the Ladinian, chirotheriid tracks and tracks referred to a crurotarsal/ dinosauroid trackmaker have been found in Iberia. However, these tracks are not useful in biostratigraphic and biochronological studies.

In sum, the Iberian record from the Anisian is coherent with the global biochronology of Triassic tetrapod tracks, but in the late Olenekian and the Ladinian the record is very scarce (Fig. 10).

Upper Triassic

For the Carnian to Rhaetian, Klein & Haubold (2007) propose two biochrons. Biochron V has a temporal range from lower Carnian to lower Norian and is composed of the ichnotaxa Atreipus, Grallator and Brachychirotherium (Klein & Haubold, 2007); biochron VI, ranging from the middle Norian to Rhaetian, consists of Grallator, Eubrontes Hitchcock, 1845 and Brachychirotherium (Klein & Haubold, 2007). By contrast, Klein & Lucas (2010a) propose the Brachychirotherium biochron for almost all the Late Triassic (from lowermost Carnian to Rhaetian). This biochron is composed of the assemblage comprising Brachychirotherium, Atreipus, Grallator, Eubrontes, Apatopus, Rhynchosauroides and dicynodont tracks (Klein & Lucas, 2010a).

In the Iberian Peninsula there are only two localities in the Upper Triassic. Pérez-López (1993) classified a trackway found in the Keuper facies as Brachychirotherium cf. gallicum. In Europe this facies spans from the late Middle Triassic (Ladinian) through the entire Late Triassic (Carnian to Rhaetian) (Sues & Fraser, 2010). The presence of Brachychirotherium is typical of the lowermost Carnian–Rhaetian, and this could be the age of these Spanish tracks. The other tracksite from the Upper Triassic presents Eubrontes and Anchisauripus and is dated as Rhaetian in age (Pascual-Arribas & Latorre-Macarrón, 2000).The ichnotaxon Eubrontes is typical of biochron VI (early Norian–Rhaetian) of Klein & Haubold (2007) and the Brachychirotherium biochron (lowermost Carnian–Rhaetian) of Klein & Lucas (2010a). Although the Iberian record for the Upper Triassic is not abundant, the data on these tracks are consistent with the global biochronology of Triassic tetrapod tracks (Fig. 10).

Tetrapod and track diversity in the Triassic of Iberian Peninsula

A noteworthy point that emerges from the previous review is the high difference in ichnodiversity among the Triassic stages in the Iberian Peninsula. According to Díaz-Martínez, García-Ortiz & Pérez-Lorente (2015), this difference can be explained in at least three ways. The first explanation would be that this is a consequence of a greater diversity of trackmakers in a concrete age than in others, this diversity being reflected in the track record. It is also possible that in one age there were more suitable facies for preserving the tracks, so although the diversity might in fact be similar in all the ages, in the Anisian it seems highest; there would thus be a preservational bias against the other ages. Finally, the high ichnodiversity could also be explained by weathering and erosion processes that affect the rock outcrops as well as the exposed surface area of the track-bearing layers.

The poor track record during Olenekian (Early Triassic) in Iberia is also observed in other European regions. It has been explained as a product of both an ecological bias (only coastal dwellers would be potentially recorded) and/or a real “evolutionary pattern” due to a slow recovery in diversity from the previous Permo-Triassic mass extinction (Avanzini, Bernardi & Nicosia, 2011). The high diversity of the tetrapod track record during the Early-Middle Triassic could be related with the radiation of tetrapods, reflecting the morphological diversity spanning from a stem-reptile to a “mammalian” foot, from a basal crocodilomorph to a dinosauromorph foot (Avanzini, Bernardi & Nicosia, 2011). The herein presented data indicates the Anisian as being the age with highest ichnodiversity. As in other European regions the decrease in the tetrapod tracks occurrences after the Anisian could be largely related with a great rise of the sea level and the consequent change to marine environments (e.g., Avanzini, Bernardi & Nicosia, 2011; Fortuny et al., 2011). In spite of a probable influence of facies bias, the pattern of the Iberian record is consistent with those observed in Germany, France, Italy and USA (Hunt & Lucas, 2007b; Avanzini, Bernardi & Nicosia, 2011), suggesting that the vertebrate track record reflect an evolutionary pattern. As previously stated (e.g., Avanzini, Bernardi & Nicosia, 2011) the track reliability for evolutionary studies is confirmed.

Global track record is much more abundant than the skeletal record and provides data as reliable as those obtained from skeletal remains (Carrano & Wilson, 2001; Avanzini, Bernardi & Nicosia, 2011). This fact is also relevant in the Iberian Triassic record. The skeletal remains of tetrapods, excluding marine groups, from the Triassic of the Iberian Peninsula are rather scarce (see Fortuny et al., 2011). In the Anisian, capitosaurs, archosauriforms, procolophonids and mastodontosaurid stereospondyls have been found in Catalonia (e.g., Gaete et al., 1996; Fortuny et al., 2011; Fortuny et al., 2014). Phytosaurs, metoposaurid temnospondyls have been identified in the Carnian-early Norian of Portugal (Steyer et al., 2011; Mateus et al., 2014). An indeterminate temnospondyl has been cited from Late Triassic of Aragón (Spain) (Knoll, López-Antoñanzas & Molina-Anadón, 2004). Finally, a mastodonsaurid stereospondyl and the temnospondyl Metoposaurus algarvensis have been found in the Triassic-Jurassic boundary of Portugal (Witzmann & Gassner, 2008; Brusatte et al., 2015). Of all the skeletal remains found in the Triassic of the Iberian Peninsula, only the Anisian archosauriforms and procolophonids can be considered as the probably trackmakers of the chirotheriid and Procolophonichnium tracks of the same age. Therefore, the track diversity increase and complement the skeletal record to a better understanding of the Triassic tetrapod diversity in the Iberian Peninsula.

In order to have a more complete vision of the Triassic track record in the Iberian Peninsula, it is therefore important to reassess the rest of the Triassic Iberian ichnological localities not included here because these do not yet have a concrete temporal geological context.

Conclusions

The ichnotaxonomy of historic vertebrate tracks found in two sandy slabs from the Anisian (Middle Triassic) of the Moncayo massif (Iberian Range, NE Spain) has been re-studied. The tracks previously considered Chirotherium ibericus by Navás, and other tracks of the same shape found in the two slabs, have been reassessed and have been classified as Chirotherium barthii. Chirotherium ibericus has been deemed to be a junior synonym of Chirotherium barthii. The rest of the studied tracks have been assigned to Chirotheriidae indet., Rhynchosauroides isp. and undetermined material. All the tracks classified as Chirotherium barthii in the Iberian Peninsula are characterized by their small size. This point and other reports of small-sized C. barthii in other localities around the world shed new light on the differentiation between small-sized C. barthii and C. sickleri. The C. barthii-Rhynchosauroides ichnoassemblage present in the Navás tracksite (Anisian in age) is typical of biochron II or the Chirotherium barthii biochron, of an Olenekian-lower Anisian age. This ichnoassemblage has also been found in other coeval Iberian localities. Although the Iberian record of Triassic tracks is not continuous and in some ages is more abundant than others, in general it is coherent with the global biochronology of Triassic tetrapod tracks. This further corroborates the usefulness of vertebrate Triassic tracks in biochronology. In the lowermost Lower Triassic-upper Lower Triassic interval, the record is very scarce and only the ichnotaxon Rhynchosauroides is cited. The record for the uppermost Lower Triassic-Middle Triassic is abundant. The most complete record is the ichnoassemblage from the Anisian, which is composed of Dicynodontipus, Procolophonichnium, Rhynchosauroides, Rotodactylus, Chirotherium, Isochirotherium, Coelurosaurichnus, and Paratrisauropus. The late Olenekian and Ladinian record is not well represented. Finally, Eubrontes, Anchisauripus and probably Brachychirotherium have been identified although the Iberian record for the Upper Triassic is not abundant. The analysis could be more complete if the whole of the Iberian record were analyzed. With this paper, therefore, we emphasize the need to reassess the Triassic vertebrate track record of the Iberian Peninsula and specify the age of the localities, in order to have a complete image of this record and compare it with the tetrapod-track-based biochronology and biostratigraphy. Triassic skeletal remains are scarce in the Iberian Peninsula when compared with the ichnological record. Therefore, the track diversity shown in this paper throughout the Triassic complements and improves the information about the tetrapod diversity in the Iberian Peninsula for this age.

Supplemental Information

Table S1 Summary of all the Iberian Triassic tracks published in the Iberian Peninsula

Click here for additional data file.

Our thanks go to Juan Jesús Bastero for providing the information about the discovery and the history of fossil. The “Museo de Ciencias Naturales de la Universidad de Zaragoza” and the “Colegio del Salvador (Jesuitas), Zaragoza” permitted us to study and photograph the specimen. We thank Ester Díaz-Berenguer (Museo de Ciencias Naturales de la Universidad de Zaragoza) for allowing us to see the studied material. We thank Adán Pérez-García and Penélope Cruzado-Caballero for their comments on an early version of the manuscript. Finally, we acknowledge Spencer G. Lucas, Lida Xing and the Academic Editor Kenneth De Baets for their helpful reviews. Rupert Glasgow revised the translation of the text into English.

Additional Information and Declarations

Competing Interests

Author Contributions

The authors declare there are no competing interests.

Ignacio Díaz-Martínez, Diego Castanera, José Manuel Gasca and José Ignacio Canudo conceived and designed the experiments, performed the experiments, analyzed the data, contributed reagents/materials/analysis tools, wrote the paper, prepared figures and/or tables, reviewed drafts of the paper.

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
