# Peer review of "A reappraisal of the Middle Triassic chirotheriid Chirotherium ibericus Navás, 1906 (Iberian Range NE Spain), with comments on the Triassic tetrapod track biochronology of the Iberian Peninsula"

_PeerJ, doi:10.7717/peerj.1044_

## Round 0.1 · original submission · Minor Revisions

Thank you for submitting your work to PeerJ. The manuscript is in a good state as attested by two positive reviews and my own personal assessment (minor revisions). Two important points should be implemented during the revision:

Highlighting the biological importance: PeerJ is focused on biological sciences, so that some more explanations of the importance of your research from a (paleo)biological perspective would be in order. This could be easily done in introduction by writing some lines on importance of disentangling the biozonation of trace fossils for evolution, particularly in the Triassic context. You mention it between the lines as you mention that the high ichnodiversity might also be related with evolution of the trace producers or even contain a biogeographic signal. Biozonation is a more geological application, but is key component to be able to answer paleobiological questions like “Does ichnofossil diversity related with diversity of their trace makers ?” and should therefore be highlighted.

History of Chirotherium ibericus: It is a bit confusing as the first lines describe the history of Longinos Navás, the person who discovered these traces, which is not all I think this part should be shorted relevant to the “history of Chirotherium ibericus”. It should start by introducing the traces and subsequently one could say something about its discoverer. It would be of interest also to say how the trace producer might have looked like. You briefly touch this in line 115, but you do not discuss the recent views on this topic.

In addition to the comments by the reviewers, the following points needs to be addressed in the revision:

Line 37: please change to “wide distribution across the supercontinent Pangea” (adding the term Pangea for completeness sake)
Line 37-38: As PeerJ is oriented at biological sciences in general, it would be welcome to slightly explain what trace fossils are and why they are important, particularly from a biological perspective as well as more specifically for the Lower Triassic (see point 1)
Line 49: please explain what you mean by “extramorphological variations” as not all readers might be familiar with this term
Line 88-122: Please this paragraphs; you could start with introducing the discovery of Chirotherium ibericus and later go into the details of its discoverer. Furthermore, the part on Longinos Navás could be shortened as they are not really relevant for the history of Chirotherium ibericus. What would be interesting to elaborate on is the possible trackmakers. You discuss that Navàs proposed an amphibian as trackmaker, but do not discuss the current view(s) on this. This would be quite relevant, also for the other traces, if you want to compare them with the hypothesis that diversity of these groups was higher during time the ichnoassemblages were also more diverse or not ?
Line 142-145: the age assignment seems to Permian or Triassic are quite coarse, which is not surprising in these types of environments. Could please briefly mention on what these age assignments are based on (biostratigraphy, palynology, lithostratigraphy, etc.). “attributed to Permian based on …” or “considered Anisian in age based on ….”
Line 174: please add manus (forelimb) and pes (hindlimb) tracks or something similar for persons not familiar with this terminology
Line 452-453: If I understand it correctly you mean that these tracks now seem to be tridactyl and mesaxonic, but they were probably more complex and differently shaped before fossilization? This sentence is a bit confusing, maybe rephrase.
Line 480-481: I think the habitat and rate of evolution do not only differ between groups, but also within taxa
Line 616-620: You specifically discuss the high ichnodiversity in the Anisian for the Iberian Peninsula, but are similar pattern seen at other sites. These possible explanations you give are interesting and at least some of them could also be tested by looking at this more globally. Are the suspected trace makers more diverse during this time, which could certainly by tested by compiling data from the literature? Have bones of possible tracemakers been found at all in the Iberian Peninsula ?
Line 623-624: It could potentially not only be related with weathering and erosion, but also with exposed surface area of the ichnofossil-bearing layers. The latter could make it easier to discover tracks.
Line 666: Do you really mean “lefting”. Sounds odd.

·

Basic reporting

This is an important contribution because these Middle Triassic chirotheriid tracks show the global biochronological features. Also, it’s very nice to see any old specimens have been reviewed.

In China, we have some new records need to be added to this MS for the global contrast.

Xing, L.D., Klein, H., Lockley, M.G., Kan, Z.Z., Zhang, J.P., Peng, G.Z., Ye, Y., 2014. First chirothere and possible grallatorid footprint assemblage from the Upper Triassic Baoding Formation of Sichuan Province, southwestern China. Palaeogeography, Palaeoclimatology, Palaeoecology 412: 169–176.

Experimental design

The 3D photogrammetric images will be very helpful for the readers, to show more clearly details. Such as the Figs. 4 and 5.

Validity of the findings

No Comments

Additional comments

This is an important contribution because these Middle Triassic chirotheriid tracks show the global biochronological features. Also, it’s very nice to see any old specimens have been reviewed.
Have some small comments:
1, The Fig 1A needs a scale bar.
2, 3D photogrammetric images will be very helpful for the readers, to show more clearly details. Such as the Figs. 4 and 5.
3, In China, we have some new records need to be added to this MS for the global contrast.
Xing, L.D., Klein, H., Lockley, M.G., Kan, Z.Z., Zhang, J.P., Peng, G.Z., Ye, Y., 2014. First chirothere and possible grallatorid footprint assemblage from the Upper Triassic Baoding Formation of Sichuan Province, southwestern China. Palaeogeography, Palaeoclimatology, Palaeoecology 412: 169–176.

·

Basic reporting

No comments.

Experimental design

No comments

Validity of the findings

No comments--only minor fixes needed (see below)

Additional comments

This is a well-executed article that presents a valuable update and review of the Spanish Triassic tetrapod footprint record. As such, it makes a substantial contribution to our understanding of the Triassic tetrapod footprint record and the use of Triassic footprints in biochronology. It merits publication after minor revision.

Spencer G. Lucas

I have indicated a few edits on the manuscript (pdf attached), and numbers on the manuscript are keyed to these suggestions:
1. Given that all the specimen numbers are listed in the systematic ichnology, I see no reason to also list them here. Simply list the collection repositories where specimens were studied.
2. Lucas et al. (2010, NMMNH Bulletin 47) and Hunt & Lucas (2007, NMMNH Bulletin 41, p. 72) discussed the ichnotaxonomy of Rhynchosauroides species and made a few suggestions—these should be cited.
3. Why aren’t these chirothere undertracks? Are the skin impressions the same in the chirothere tracks and the nondescript tracks? Please discuss.
4. Rhynchosauroides is very common in the Late Triassic; this should be mentioned.
5. Klein and Lucas noted that all Brachychirotherium reports older than Late Triassic need to be re-evaluated. Also, note that Lucas and Heckert (2011, Ichnos) suggested aetosaurs (only known from Late Triassic) were the Brachychirotherium trackmakers, so this implies a Late Triassic age for Brachychirotherium records.

---

## Round 0.2 · Minor Revisions

Thank you for integrating all the suggested changes made by the reviewers and myself. Your article is as good as accepted. I still found some minor issues related with language or references (see annoted pdf). These need to be resolved before acceptance as these changes cannot be made after the paper has been officially accepted. I apologize for the inconvenience and thank you for your understanding.

---

## Round 0.3 · accepted · Accept

Thank you for integrating these last changes and adding the missing references.